# Environmental Substances Associated with Osteoporosis–A Scoping Review

**DOI:** 10.3390/ijerph18020738

**Published:** 2021-01-16

**Authors:** Hanna Elonheimo, Rosa Lange, Hanna Tolonen, Marike Kolossa-Gehring

**Affiliations:** 1Department of Public Health Solutions, Finnish Institute for Health and Welfare (THL), 00271 Helsinki, Finland; hanna.tolonen@thl.fi; 2German Environment Agency (UBA), 14195 Berlin, Germany; rosa.lange@uba.de (R.L.); marike.kolossa@uba.de (M.K.-G.)

**Keywords:** osteoporosis, chemical exposure, cadmium (Cd), lead (Pb), phthalates, per- and poly-fluoroalkyl substances (PFASs), HBM4EU

## Abstract

Introduction: Osteoporosis is a disease having adverse effects on bone health and causing fragility fractures. Osteoporosis affects approximately 200 million people worldwide, and nearly 9 million fractures occur annually. Evidence exists that, in addition to traditional risk factors, certain environmental substances may increase the risk of osteoporosis. Methods: The European Human Biomonitoring Initiative (HBM4EU) is a joint program coordinating and advancing human biomonitoring in Europe. HBM4EU investigates citizens’ exposure to several environmental substances and their plausible health effects aiming to contribute to policymaking. In HBM4EU, 18 priority substances or substance groups were selected. For each, a scoping document was prepared summarizing existing knowledge and health effects. This scoping review is based on these chemical-specific scoping documents and complementary literature review. Results: A possible link between osteoporosis and the body burden of heavy metals, such as cadmium (Cd) and lead (Pb), and industrial chemicals such as phthalates and per- and poly-fluoroalkyl substances (PFASs) was identified. Conclusions: Evidence shows that environmental substances may be related to osteoporosis as an adverse health effect. Nevertheless, more epidemiological research on the relationship between health effects and exposure to these chemicals is needed. Study results are incoherent, and pervasive epidemiological studies regarding the chemical exposure are lacking.

## 1. Introduction

The World Health Organization (WHO) defines osteoporosis as a systemic skeletal disease including reduced bone mass and micro architectural degrading of bone tissue, which causes an increase in bone frailty and susceptibility to fractures [1]. Women suffer from osteoporosis more commonly than men. Women start losing bone mass younger, have a fourfold higher rate of osteoporosis and a twofold higher rate of osteopenia at the age of 50 or above, and have fractures 5–10 years younger than men [2]. However, the density and trabecular architecture of bones are equivalent in both genders. The fracture rates in men are lower than in women, mainly because men lose less porous (trabecular) bone compared to women [3]. The risk of fractures can be assessed with a variety of techniques. Two prominent categories exist: clinical assessment of risk factors and physical measurement of skeletal mass. Various different types of bone mineral density (BMD) assessments can be applied. The most common method is dual-energy X-ray absorptiometry (DXA). Quantitative ultrasound (QUS) is regarded as a possible non-invasive alternative using sound waves to examine the velocity, attenuation, or reflection of ultrasound in the bone. Ionizing radiation is not needed, and ultrasound gives information concerning the structural organization of bone in addition to bone mass or density [1].

A total of 200 million people throughout the world are estimated to be affected by osteoporosis and furthermore, more than 8.9 million fractures take place annually. Of the global fractures, one-third takes place in Europe [4,5,6]. The osteoporosis prevalence in the European Union (EU) (based on the 27 countries) was estimated at 27.6 million in 2010 with 22 million being women and 5.6 million men. At the same time period, 3.5 million new fragility fractures occurred [6]. In the EU, the economic burden of osteoporosis and former fragility fractures was approximately €37 billion in 2010. Of this cost, incident fractures accounted for 66%, fracture care for long-term 29%, and pharmacologic prevention 5%. Prior and incident fractures caused 1,180,000 quality-adjusted life years (QALYs) lost in 2010. These costs are predicted to increase by 25% in 2025 [6].

Vulnerability to fractures is heightened by high age, menopause at an early age, a maternal incident of hip fracture, a fracture after the age of 40 years, low body weight levels, or particular diseases and treatments [4]. Ensuring a nutritious diet with sufficient calcium and vitamin D consumption, engaging in steady weight-bearing activities and refraining from under-nutrition, smoking, and heavy drinking are the cornerstones of building and maintaining healthy bone mass [7]. Also, environmental substances such as Cd, Pb, phthalates, and PFASs are expected to have adverse effects on BMD and therefore increase the risk of osteoporosis [8]. In this study, only the substances showing the strongest epidemiological evidence on humans according to the current search are selected, and therefore e.g., heavy metals such as arsenic (As) and mercury (Hg) are excluded.

This scoping review aims to present an overview of the recent research evidence of the selected environmental substances and their possible associations with osteoporosis. The rationale of this review is to present a disease-oriented approach to the risks that environmental substances can post to human health, more specifically on osteoporosis. Rather than focusing on the overall health effects of the specific substances, we chose to concentrate only on one specific health outcome, osteoporosis. This kind of approach method can be informative and useful when aiming to enhance public health and in finding solutions to tackle the increasing burden of bone health disorders. According to our knowledge, this kind of scoping review has not been conducted earlier.

## 2. Materials and Methods

This study was conducted in the supporting structure of the HBM4EU. HBM4EU is a combined effort of 30 countries and the European Environment Agency (EEA), co-funded under the Commission’s funding program Horizon 2020. The project runs from 2017 to 2021 with the aim of generating evidence of the citizens’ exposure to chemicals and on its plausible effects on health in order to strengthen policies [9].

The selection of substances to be studied under HBM4EU was conducted through a prioritization process in consultation with policymakers, scientists, and stakeholders to establish knowledge needs [10]. In total, eighteen substances and substance groups were identified in two prioritization rounds: acrylamide, anilines, aprotic solvents, As, benzophenones (UV filters), bisphenols, Cd, chromium VI (Cr VI), flame retardants (FRs), Pb, Hg, mycotoxins, PFASs, pesticides, phthalates and Hexamoll ^®^Dinch, polycyclic aromatic hydrocarbons (PAHs), chemical mixtures and emerging chemicals.

Under HBM4EU, substance-specific scoping documents were prepared including recent information about characteristics of the substances and their policy relevance [8]. Known or suspected adverse health effects were mentioned in these documents, usually without further details. These individual scoping documents were used as background material for this scoping review. For each priority substance, excluding chemical mixtures and emerging chemicals, supplementary literature searches in PubMed were conducted. Used key words were “osteoporosis”, “environmental chemicals”, “chemical exposure” and each of the priority substances and substance groups. The research evidence is based on epidemiological studies. In vivo and in vitro animal toxicity studies were not considered. Only epidemiological studies showing a possible connection between osteoporosis and the chemicals were included, and if encountered, totally opposing evidence was omitted from this review. The main focus was to include studies published after the year 2000. However, in some occasions, older studies were accepted, if this was justified based on the research question and the scarcity of recent studies.

Through the HBM4EU scoping documents and supplementary literature search, Cd, Pb, phthalates, and PFASs were identified to have an association with osteoporosis and were included to this scoping review. There are suspected associations of osteoporosis and benzophenones (UV filters), bisphenols, PAHs, and Hg, but the evidence is unclear and therefore, these substances were omitted.

Since this is a scoping review aiming to contribute an overview of the latest research evidence of the association between chemical exposure and osteoporosis without a systematic summary of the findings [11], no systematic review methods have been used.

## 3. Results

### 3.1. Cadmium (Cd)

Cadmium is a heavy metal, which is essentially found in the environment at low levels [12]. It causes toxic effects on kidneys, liver, skeletal, respiratory, and reproductive systems and is regarded as a human carcinogen. Food and drinking water are considered as main roots of origin of Cd exposure in the generic non-smoking population (Table 1). Some plants, such as rice, mushrooms, and tobacco tend to store Cd from the soil [8,12,13]. Smoking and inhaling tobacco smoke poses a significant risk of Cd exposure, and the average urinary excretion and blood concentration of Cd are twice as emphasized in smokers than in non-smokers [8,12,14]. Vulnerable populations include smokers, pregnant, and post-menopausal women as well as elderly and children [8,13,14]. Long-term exposure to Cd can be determined from urine samples, while short-term exposure can be seen in blood (Table 2).

Kidneys are a target organ for Cd toxicity after chronic exposure, and evidence exists of dose and response relations for effects on kidneys. Cd can accumulate in the cortex of the kidney, especially in the proximal renal tubular cells, causing nephric tubular dysfunction [12,13,14,24]. European Food Safety Authority (EFSA) [25] conducted a meta-analysis investigating the associations between urinary Cd (U-Cd) and β2-microglobulin (β2M) as an indicator of early nephrotoxic effects. The goal was to derive a benchmark portion and its 95% confidence lower threshold (BMDL95) for humans, applying cut off points essential to clinical changes in the target organ. A benchmark concentration was calculated, which is suitable to prevent an excretion of β2M that is elevated by 5% and selecting the 95% lower confidence limit for this value (BMD5L95). For the general population, this resulted in a concentration of 4 µg U-Cd/g creatinine. Bone effects may take place at exposure levels corresponding with those causing kidney effects to elevate, and exposures leading to Cd levels in urine of 5 µg g^−1^ creatinine (5 nmol mmol^−1^ creatinine) or more (or blood Cd of 5 µg L^−1^ or more) aggravate the risk of bone effects [12]. Cd urine levels in population not exposed are normally less than 0.5 µg/g creatinine and values of more than 1–2 µg/g refer to exposure or heightened body burden. The corresponding blood Cd levels are typically lower than 0.5 µg/L and values of more than 1.0 µg/L refer to exposure. Furthermore, blood levels above 5 µg/L are regarded as hazardous [26].

In humans, the biological half-life of Cd is 10–35 years [27]. There are plenty of epidemiological studies conducted on the relations between Cd exposure and effects on health in general. Damage to bone demineralization and fractures can take place caused by prolonged exposure to Cd [14,24]. Extreme exposure to Cd may lead to itai-itai disease, which causes osteomalacia and/or osteoporosis with several fractures [14]. Up-to-date cross-sectional and prospective studies have demonstrated negative associations between exposure to Cd and low mineral density, as well as an elevated risk of osteoporosis [28,29,30,31]. Cd concentrations in urine have been positively linked with osteoporosis with a significant negative association with BMD. U-Cd has been inversely linked with BMD at the total body, femoral neck and volumetric femoral neck, total hip, and lumbar spine. Clinical trials have proven that Cd poisoning gives rise to osteoporosis as well as heightens fracture risk [30,31] (Table 3).

Engström et al. [30] adjusted the association between dietary Cd exposure and the outcomes of BMD, risk of osteoporosis and any first fracture for age, body mass index (BMI), consumption of post-menopausal hormones, total physical activity levels, smoking, alcohol use, inflammatory joint conditions and dietary issues such as calcium intake. Wallin et al. [31] included potential risk factors such as age, smoking, BMI, and physical activity levels in their models when examining Cd exposure in relation to BMD and incident fractures. In some of the studies of the review by Åkesson et al. [28], adjustment or stratification was conducted for smoking when examining the harmful effects of Cd exposure on bone health. The authors concluded that bone effects can be confounded by smoking, since tobacco smoke, besides being an evident risk factor for osteoporosis, can include other agents that affect BMD and fracture risk. Tobacco is known to form a significant root of Cd exposure [8]. Lv et al. [29] used gender, age, smoking status and BMI as covariates, and mentioned that these confounders could have affected the results of association between osteoporosis and long-lasting environmental Cd exposure deriving from diet. They recognized the fact that osteoporosis is age- and gender-related disease with low BMI also causing a risk factor.

### 3.2. Lead (Pb)

Lead is a toxic metal that is essentially present in the Earth (Table 1). Pb has been widely used, which has caused vast contamination in the environment, exposure to humans, and considerable public health issues all over the world. As a cumulative toxicant, it affects several body systems and is especially harmful to young children. Pb is distributed to the brain, liver, kidney, and bones, and it is stored and accumulated in the teeth and bones. Since Pb stored in the bone can remobilize into the blood during pregnancy, the fetuses are at risk to be exposed. The occupational groups with exposure to Pb are also among the vulnerable population [40]. Recent exposure to Pb can be determined from blood while long-term exposure is usually determined either from urine or bone (Table 2). The half-life of Pb in blood and bone are approximately 30 days and 10–30 years, respectively, excretion being mainly through urine and feces [41].

Exposure to Pb may be a risk factor for osteoporosis development, even though epidemiological studies on this linkage are limited and inconsistent. The study using data from NHANES (*N* = 4689; ≥50 years of age) presented a prominent inverse association between exposure to Pb and BMD, but among white study subjects only. Nevertheless, due to the cross-sectional study design, they couldn’t make any statements about the temporal order of the association [32]. According to another NHANES-based study (*N* = 2981 women), Pb may interact with other factors during post-menopausal osteoporosis to provoke osteoporosis since Pb prevents activation of vitamin D, taking in dietary calcium and multiple regulatory factors of cell function in bone [33]. Also, occupational Pb exposure seems to be associated with decreased BMD and osteoporosis prevalence, with U-Pb having a closer relationship with Pb-associated osteoporosis than B-Pb [21]. Studies using the NHANES III survey dataset of nearly 3000 women found that Pb stored in bone may greatly cause accumulation of Pb levels in blood in perimenopausal women due to post-menopausal bone mineral resorption. BMD was markedly inversely associated with Pb level in blood after modification for other factors generally associated with blood Pb (such as age, ethnicity, residence, income, smoking, and use of alcohol). A low BMD measurement may express either recent decrease in BMD or longtime bone mineral status, and it is feasible that Pb may influence BMD rather than BMD’s influencing on Pb [34]. Some clinical studies have shown that Pb accumulated in the body has negative impacts on bone, decreases cortical width and bone density and consequently increases risk of fracture. Also, greater dividing of Pb into blood versus bone in subjects with weaker bone structure and lower volumetric bone densities was detected, indicating that releasing of Pb from bone in women in post-menopausal stage may be a relevant issue of concern [35] (Table 3).

Campbell and Auinger [32] used covariates, which are proven to be associated with bone density, such as age, BMI, intake of calcium, use of alcohol, smoking, medication, levels of physical activity, menopausal stage (post-menopausal subjects only included) and even socioeconomic status in their analyses of the association between Pb exposure and osteoporosis. Sun et al. [21] acknowledged the traditional risk factors behind osteoporosis such as sex, age, hormones, poor nutrition, insufficient physical activity, calcium and vitamin D deficiencies, smoking and heavy alcohol use. The characteristics of the study subjects were examined, e.g., age, height, and weight (BMI), medical history, drug use, smoking, and alcohol use and physical exercise. The association between occupational exposure to Pb and low bone mass was investigated, and there were no marked discrepancies in lifestyle, nutrition, smoking, or alcohol use between the exposure and control group. Nash et al. [34] tested the presumption whether bone break down in post menopause aggravates blood Pb-levels, and the results were adjusted for some of the traditional risk factors of both blood Pb and osteoporosis such as age, race/ethnicity, smoking and alcohol use. Menopausal stage was also added to the model. Wong et al. [35] measured bone Pb of the post-menopausal women, and they adjusted the results for age, BMI, diabetes, and antiresorptive therapy (used for increasing bone strength in osteoporosis). Medical history including e.g., calcium and vitamin D intake, use of medication, and incident fragility fractures was collected from the database of CaMos-study (Canadian Multicenter Osteoporosis Study), and it was used to describe the bone characteristics of the study subjects. This specific background information wasn’t applied in assessing the correlations of Pb-levels with bone health.

### 3.3. Phthalates

Phthalates (or phthalate esters), more specifically ortho-phthalates and the non-phthalate substitute Hexamoll^®^ DINCH, form a group of plasticizers manufactured in massive amounts. Consequently, they are the most consumed plasticizers globally. They are present all over in the environment [15], and exposure occurs through multiple sources (e.g., food, water, air and consumer products) (Table 1). Ortho-phthalates, made out of alcohols, are divided according to their molecular structure into either low or high molecular weight ortho-phthalates: LMW or HMW. Phthalates are known to result in various harmful effects in humans especially on the endocrine and reproductive systems. Children, fetuses and thus, pregnant women are the most vulnerable groups [8,16]. Phthalates are quickly metabolized and most of them are excreted in urine within 24 h [16,19] (Table 2).

There is a limited knowledge of the association of phthalates to human bone health. The results of epidemiological studies indicate that phthalate exposure may adversely affect bone homeostasis and BMD and increase osteoporosis risk. An increasing number of studies have scrutinized the implication of endocrine-disrupting compounds (EDCs) on health, showing that exposure to phthalates leads to harmful health outcomes, including bone loss [36,37,42,43] (Table 3).

Min and Min (2014) [36] researched the link between phthalate metabolites in urine and BMD and osteoporosis in post-menopausal women, and the covariates used included e.g., race/ethnicity, menopausal causes (natural/surgical), cigarette smoking habits, moderate physical activity levels, history of diseases (fractures), BMI, and calcium intake. The results were adjusted according to the covariates.

### 3.4. Per- and Poly-Fluoroalkyl Substances (PFASs)

Per- and poly-fluoroalkyl substances are omnipresent environmental contaminants found in humans all over the world. The exposure happens via e.g., food, drinking water, and consumer products (Table 1). According to the current knowledge, the use of PFASs leads to environmental contamination and increased human body burden. Various long-chain perfluoroalkyl acids have been identified as highly perseverant, bio-accumulative and toxic–also shorter chain PFASs, used as alternatives, have been recognized as very persistent and mobile in the environment. The best-known substances of perfluorooctane sulfonate (PFOS) and perfluoro-octanoic acid (PFOA) are classified as carcinogenic, toxic to reproduction and specific target organs and acute toxic for different exposure routes. Children and occupationally exposed workers belong to vulnerable populations. PFASs are determined from blood, sometimes also from breast milk (Table 2). PFASs fasten to proteins and the elimination kinetics depends on species; humans have the longest half-lives of up to 8.5 years for perfluorohexane sulfonic acid (PFHxS) [8,17].

Serum PFAS concentrations have been linked with lower BMD varying on the basis of specific PFAS and bone sites measured in a prominent sample of the U.S. adult population (NHANES 2009–2010, *N* = 1914). In women, osteoporosis had a linkage with exposure of PFAS, referring to a small number of study cases [38]. The study using data from NHANES 2005–2008 (*N* = 2339) found that serum PFOS level was linked with reduced BMD of total lumbar spine in non-menopausal women [39] (Table 3).

Khalil et al. (2016) [38] studied the association between serum PFAS concentration and BMD and osteoporosis. They included covariates such as age, ethnicity, sex, BMI, smoking, intake of daily milk, physical activity levels, and menopause. Lin et al. (2014) [39] scrutinized the association between serum PFOA and PFOS concentrations and BMD. The included covariates were age, sex, ethnicity, smoking, BMI, alcohol use, and history of osteoporosis, fractures, and medication. These covariates were used in adjusted models.

## 4. Discussion

There is evidence that among other adverse health effects of environmental substances, the effects on human bones are also occurring. The evidence suggests that exposure to Cd, Pb, phthalates, and PFASs can cause alterations to bone metabolism and can lead to osteoporosis. Also, according to the literature, the substances such as bisphenols, As, Hg, and PAHs are supposed to cause harmful effects on bone health, but the epidemiological evidence is still to be completed. BMD is widely used as an endpoint to investigate the association of chemical exposure and osteoporosis. In many studies an inverse association of chemicals and BMD is discovered [28,32,34,36,37,38,39,44]. Women, especially at perimenopausal and post-menopausal stage, have a greater risk of adverse effects. Reasons are e.g., that total hip and femur neck BMDs are drastically reduced with age and there is an accelerated post-menopausal bone loss when the protective effect of estrogen is lost [1,7,35,36,37,39]. Nevertheless, the relation between chemical exposure and bone metabolism in humans is yet widely understudied, and thorough conclusions of the causality and temporal sequence of the associations are difficult to draw. Some studies show inconsistent and even contradictory results. The level of chemical exposure often varies between the studies, and the study sizes are sometimes too small to draw a generally valid conclusion. In order to come up with accurate conclusions, the threshold values of BMD and standardized diagnostic methods of osteoporosis should be used.

Furthermore, there is a controversy over the starting point for the bone effects and more research is needed to identify the causal relation between e.g., low-level Cd exposure and bone effects. It is hoped that this low-level exposure threshold can be specified in the future with additional data.

The combined effects of exposures to multiple chemicals are scarcely studied, but researchers have observed the interactive effects of Cd and Pb on BMD in a Chinese population living in control and polluted areas [44]. They found that people living in the polluted area had markedly higher Cd and Pb levels compared to those living in the control area, and the BMDs of women from the polluted area were significantly lower than those of women from the control area. In addition, the BMD diminished with growing values of Cd and Pb. The study supports the previous evidence by strengthening the findings that Cd and Pb may adversely influence the bone [21,45], and also demonstrated that these substances may have interactive impacts on BMD. More specifically, the relative excess risks due to interaction (RERIs) of female and male study subjects with both high levels of B-Cd and B-Pb were 0.45 and 1.16, respectively. This points out that the estimated joint effect of B-Cd and B-Pb together on the additive scale is higher compared to the sum of the estimated effects of B-Cd and B-Pb alone. Consequently, the interaction was proven positive on the additive scale [44].

In the majority of the studies investigating the possible role of chemical exposures associated with BMD and osteoporosis, the traditional risk factors of osteoporosis were used as covariates and were included in the statistical models. The authors mostly acknowledged the prevalent role of these factors in osteoporosis incidences and in interpreting the results.

Human biomonitoring (HBM) studies aim to identify and quantify chemicals and their metabolites in human biological matrices. On the grounds of these findings, the rendition of the measurements is conducted to decide whether chemicals’ management measures or regulation is integral. Evaluations can be conducted by comparing the measured HBM levels of a selected substance with either a reference value of the general population’s background body burden or with the preferred method of an internal benchmark level on the grounds of epidemiological and toxicological data [46]. The determination of reference values of environmental chemicals is somewhat challenging since there are many pathways of exposure (e.g., occupational vs. home) and there are various different stakeholders producing the guidance value information. Some examples of these organizations setting guidance values include German Human Biomonitoring Commission (HBM Commission), European Chemicals Agency (ECHA), WHO, Centers for Disease Control and Prevention (CDC), Scientific Committee on Occupational Exposure Limits (SCOEL) set up by the European Commission and EFSA [47,48,49,50,51].

In the light of safeguarding human health and the environment, the EU has significantly developed and expanded its chemicals legislation after the acceptance of the first chemicals-related directive from the late 1960s. The legislation monitors both the chemical sector and connected industries using chemicals [52].

The regulatory measurements of Pb include the procedures of phasing out of leaded gasoline in most countries; this has successfully reduced the blood Pb concentrations of populations. However, there is still work to be done regarding the phasing out of Pb paint [40]. The Drinking Water Directive (98/83/EC) of the EU has set the health limit value regarding Pb in drinking water as 10 µg/L. Based on the EU directives (2013, 2008, 2000) the Pb concentrations in inland surface waters should be reduced to a limit of 1.2 µg/L, and 1.3 µg/L in outland surface water. The directives (2008) have also determined a regulatory limit value regarding Pb in air as 0.5 µg/m^3^ per calendar year. Regulatory limit value regarding Pb in soil is 50–300 mg/kg. Pb in foodstuff is also regulated by the EU even though there has been no evidence of a threshold value for a variety of crucial outcomes such as developmental neurotoxicity and nephric impacts in adults. The Chemical Agents Directive (98/24/EC) defines regulations for occupational exposure [8].

Even today, despite heavy regulations in place, certain *phthalates* (i.e., their metabolites) are detected in the urine of nearly every person. Reprotoxic substances, such as the following phthalates: DnBP, DiBP, BBzP, DEHP, DMEP, DnPeP, DiPeP, DHNUP, DnHP, and DCHP are commonly not permitted to be put onto the EU market in substances or mixtures when concentrations limits are identical or surpass 0.3%. The placing on the market of items including DEHP, DnBP, DiBP, and BBzP in a concentration equal to or above 0.1% by weight individually or in any connection in any plasticized material is restricted. DiDP, DiNP, and DnOP are restricted in toys and childcare items that can be put into the mouth having a concentration limit of 0.1%. Many of them are subject or become subject to authorization in 2020, such as DEHP, BBzP, DiBP, DnBP, DMEP, DnPeP, and DiPeP, which means that they are not permitted to be put on the European market without authorizations. However, the consumer products from Asia and U.S. can contain phthalates as the authorization requirements do not apply to the goods which have been imported [8,53].

Of *PFASs* PFOA and PFOS are the most studied; in the EU and also elsewhere, current regulatory actions mostly involve PFOS and its derivatives. PFOA and PFOA related substances are under revision as global persistent organic pollutants (POPs). PFOS and PFOA as well as the precursory substances are subject to the EU restrictions. Strategic International Approach to International Chemicals Management (SAICM) has determined PFASs as an issue of concern [8,18].

It should be underlined, that this scoping review does not attempt to evaluate the quality of evidence or make a synthesis of the research. Therefore, the evidence gathered in this review is by no means comprehensive, and also the research on chemical exposure is rapidly evolving, which requires constant and extensive following and updating of this subject.

Even though there is evidence, identified during the literature search, that the above-mentioned chemicals and osteoporosis are interrelated, the findings are somewhat contradictory and only directional at the moment. Also, there is an evident lack of epidemiological studies on the subject. More extensive research on the relationship between health effects and exposure to these chemicals is needed. Additionally, the impacts in humans require evidence from more exhaustive longitudinal studies. People are exposed to many chemicals simultaneously, and therefore there is a prompt demand to launch studies striving for elaborating the plausible combined effects of chemicals in humans. The current issue of chemical exposure demands growing and multidisciplinary attention for the sake of protection of citizen’s health in the EU and also globally. Nevertheless, this scoping review demonstrates that there is a link between exposure of certain environmental chemicals and adverse effects on bone health and that additional research is warranted in order to protect public health.

## 5. Conclusions

There are a number of studies that suggest that:Exposure to Cd, Pb, phthalates, and PFASs may have harmful effects on BMD and may increase osteoporosis risk.In epidemiological studies, an inverse association of chemicals and BMD is identified; the higher the chemical levels in the measurement matrix, the lower the measured BMD at different bone sites.The concern of chemical exposure is rapidly growing, which demands launching of extensive epidemiological studies on the subject. Multidisciplinary and prompt action for protecting citizens in the EU and globally is required.The study results indicate that osteoporosis might be an underestimated endpoint that is not sufficiently integrated in epidemiological studies.People are increasingly exposed to environmental chemicals during their daily activities. The increasing amount of chemical exposure causes body burden and can contribute to the development of diseases. Therefore, protecting people from excessive exposure is relevant from the public health point of view.

## Figures and Tables

**Table 1 ijerph-18-00738-t001:** Sources of exposure.

Substance	Sources of Exposure
Cd [8,13,14]	❖ Transported via air, water and soil. ❖ Exposure through: diet/contaminated food and drinking water. Tobacco smoke and inhalation by workers in a range of industries. ❖ Natural and anthropogenic sources of Cd: erosion of parent rocks, volcanic eruptions and forest fires. Use in plastics as color pigment and stabilizer, automobile radiators, alkaline batteries, mining activities, fertilizers, sewage sludge, inappropriate waste disposal.
Pb [8]	❖ Exposure through: inhalation, oral and trans-placental and via direct contact with Pb products. Mostly via environment: air, water and soil. ❖ Sources: multiple man-made substances such as petrol additives. Pb-based paints. Inorganic Pb or Pb salts (Pb pipes and solders in plumbing systems, Pb-soldered cans, batteries etc.).
Phthalates [8,15,16]	❖ Ubiquitously present in the environment. ❖ Exposure through: ingestion, inhalation and dermal exposure. ❖ Sources: released, leached, migrated or evaporated into environment (water, air, dust), foodstuff or other materials (personal-care and consumer products). ❖ Sources for DEHP: contamination of food and food contact materials.
PFASs [8,17,18]	❖ Ubiquitously present in the environment. ❖ Exposure through: diet/food, ❖ Sources: diet/food (especially seafood) drinking water, consumer products (textiles, clothes, footwear, furniture and carpets), lubricants, waxes, paints, and fire-fighting foam, and indirectly through transformation of precursory substances.

**Table 2 ijerph-18-00738-t002:** From which matrix-specific substances are measured.

Substance	Matrix
Cd [8,14,19,20]	❖ Urine: long-term accumulation and exposure; spot urine sample is useful. ❖ Blood: short-term exposure. ❖ Sometimes measured from human hair, nails, saliva, breast milk, or placenta.
Pb [21,19,22,23]	❖ Blood: recent exposure; plasma samples are better than whole blood samples. ❖ Bone: long-term exposure. ❖ Urine: long-term occupational exposures; 24 h samples are preferred. ❖ Sometimes measured from human hair, nails, breast milk or placenta.
Phthalates [8,16,19,20]	❖ Urine: 24 h samples are informative and reliable for daily intake; spot samples are mostly used in population-based studies. ❖ Sometimes measured from saliva, meconium, semen or placenta.
PFASs [8,20]	❖ Blood (serum or plasma). Sometimes breast milk.

**Table 3 ijerph-18-00738-t003:** Results and the studies selected for the scoping review.

Study:	Chemical:	Population:	Study Design:	End Measurement(s):	Exposure Effect or Biomarker Concentration Measured:	Results:	Main Conclusions:
Åkesson et al. (2014) (review) [28]	Cd	Cd exposure and bone effects: 15 studies with study population from *n* = 270 to *n* = 10,978. Cd exposure and fractures: 6 studies with study population from *n* = 506 to *n* = 22,173.	Prospective, retrospective, or cross-sectional.	Bone effects or fractures. Most of the studies used DXA and defined low bone mass/osteoporosis based on the z- or T-score.	Threshold of bone effects measured mostly by U-Cd, sometimes from B-Cd. Exposure measured through biomonitoring, in two studies also dietary exposure was measured. Use of different exposure assessment methods (urine, blood and dietary) reduces the possible confounding.	In most of the studies, associations between Cd exposure and low BMD as well as an increased risk of osteoporosis were observed. In four studies no statistically significant Cd-related effect on BMD were observed.	There is an association with exposure to low concentrations of Cd and effects on bone (incl. increased risk of osteoporosis and fractures).
Lv et al. (2017) [29]	Cd	*n* = 1116 subjects	Population-based study.	BMD and the levels of urinary markers of early renal impairment.	U-Cd	U-Cd concentrations of subjects: 0.21 to 87.31 µg/g creatinine (median 3.97 µg/g). A significant negative association of U-Cd concentrations with BMD. A positive association with U-Cd concentrations with osteoporosis. The odds of osteoporosis increased with increase of U-Cd concentration.	An inverse association between the body burden of Cd and osteoporosis was observed. Toxic effect of Cd on bone may take place together with nephrotoxicity.
Engström et al. (2012) [30]	Cd	*n* = 2676 women (aged 56–69 years)	Population-based prospective cohort study.	BMD at the total body, femoral neck, and lumbar spine by using DXA. Risk of osteoporosis: hip or spine. Risk of any first incident fracture.	U-Cd	An inverse association of dietary Cd and BMD at the total body (*p* = 0.045) and the lumbar spine (*p* = 0.004). No association at the femoral neck (*p* = 0.89). When adjusting for dietary factors (calcium, magnesium, iron and fiber), the inverse association became more pronounced. Comparison of high dietary Cd exposure (≥13 µg/day, median) with lower exposures (˂13 µg/day) resulted in a 32% increased risk of osteoporosis (95% CI: 2–71%) and 31% increased risk for any first incident fracture (95% CI: 2–69%). Comparison of high dietary Cd with high U-Cd (≥0.50 µg/g creatinine) among never-smokers resulted in OR = 2.65 (1.43–4.91) for osteoporosis and OR = 3.05 (1.66–5.59) for fractures.	Even low-level Cd exposure from food was associated with low BMD and an increased risk of osteoporosis and fractures.
Wallin et al. (2016) [31]	Cd	*n* = 936 men (aged 70–81 years)	Prospective cohort study.	BMD at the total body, hip, and lumbar spine by using DXA. Incident fractures.	U-Cd	Significant negative associations between U-Cd and BMD were observed; lower BMD (4% to 8%) in all sites was detected for in the fourth quartile of U-Cd. Positive associations between U-Cd and incident fractures, especially non-vertebral osteoporosis fractures in the fourth quartile of U-Cd was observed. U-Cd was significantly associated with non-vertebral osteoporosis fractures (adjusted hazard ratio 1.3 to 1.4 per µg creatinine) also among never-smokers, but not with the other fracture groups.	Among elderly men, relatively low Cd exposure through diet and smoking was associated with increased risk of low BMD and osteoporosis-related fractures.
Campbell & Auinger (2007) [32]	Pb	About 40,000 people (≥2 months of age). Analysis included subjects ≥50 years of age (*n* = 8654) and the final analysis 4689 subjects from NHANES III-survey.	A secondary analysis of a cross-sectional study.	Primary outcome: BMD of the total hip measured by DXA. Clinical outcomes: the presence of back pain and history of osteoporotic-related fracture.	B-Pb	The adjusted mean total hip BMD: Non-Hispanic white males with a blood Pb-level in the lowest tercile versus the highest tercile was 0.961 g/cm^2^ and 0.934 g/cm^2^, respectively (*p* ˂ 0.05); White females with marginally significant difference (0.05 ˂ *p* ˂ 0.10 in comparison to lowest tercile).	Among white subjects, significant inverse association between Pb exposure and BMD was detected. Between blood Pb-level tercile and clinical outcomes no associations were observed.
Silbergeld et al. (1988) [33]	Pb	2981 women (both black and white) from NHANES II-survey.	-	Pb-status in women, both before and after menopause.	B-Pb (both whole blood and plasma).	After menopause, a highly significant increase in whole blood and calculated plasma Pb concentrations was detected.	Bone Pb is not an inert storage site for Pb. Pb may interact with other factors within post-menopausal osteoporosis aggravating the course of disease by inhibiting activation of vitamin D, uptake of dietary calcium and several regulatory aspects of bone cell function.
Sun et al. (2008) [21]	Pb	*n* = 249 (191 males and 58 females).	-	BMD measured by monophoton absorptiometry. Osteoporosis defined by Z-score (Z score ˂ −2).	U-Pb and B-Pb.	In both genders, a significant decrease in the groups of the high U-Pb-level compared with the low U-Pb-level was observed. No significant difference between B-Pb and BMD was detected. The prevalence of osteoporosis increased significantly with the increase of both U-Pb and B-Pb. A dose-response relationship between Pb exposure and prevalence of osteoporosis was observed.	U-Pb had closer association with osteoporosis caused by Pb in comparison to B-Pb. Occupational Pb exposure was associated with osteoporosis.
Nash et al. (2004) [34]	Pb	2575 women aged 40–59 years, and the final analysis on 1914 subjects from NHANES III-survey.	Cross-sectional design.	BMD measured in five regions of the femur by DXA.	B-Pb.	A significant inverse relationship between BMD and B-Pb level that remained even after adjusting for other factors traditionally associated with B-Pb. A one-unit change in BMD resulted in 0.6-µg/dL lower geometric mean B-Pb level. The association remained after adjusting for menopausal status.	Among perimenopausal women, due to post-menopausal bone mineral resorption, Pb stored in bone may significantly increase B-Pb levels.
Wong et al. (2015) [35]	Pb	*n* = 38, post-menopausal women (mean age 76 +/−8).	A cross-sectional observational cohort study.	Volumetric BMD and structural parameters obtained from peripheral quantitative computed tomography images.	B-Pb (whole blood) and XRF scan to obtain bone Pb content at the mid-tibia and calcaneus. Blood Pb and bone Pb were expressed as a blood:bone Pb partition coefficient (PBB) (blood-to-bone).	Higher amounts of bone Pb at the tibia were associated with thinner distal tibia cortices (−0.972, (−1.882–0.061) per 100 µg Pb/g of bone mineral) and integral volumetric BMD (−3.05 (−6.05–0.05) per µg Pb/g of bone mineral. A higher PBB was associated with larger trabecular separation (0.115 (0.053, 0.178), lower trabecular volumetric BMD (−26.83 (−50.37, −3.29) and trabecular number (−0.08 (−0.14, −0.02), per 100 µg Pb/g of bone mineral. Total Pb exposure activities significantly related to bone Pb at the calcaneus (8.29 (0.11–16.48)).	Pb accumulation in bone can have a small harmful effect on bone structure. Greater partitioning of Pb in blood versus bone manifested more dramatic effects on both microstructure and volumetric BMD.
Min & Min (2014) [36]	Phthalates	*n* = 398, post-menopausal women ≥50 years from from NHANES-surveys (2005–2006 and 2007–2008).	-	Total hip and femur neck BMD measured by DXA and osteoporosis defined by the WHO criteria.	U-phthalate metabolites (11 different).	Increasing of the urinary mono-*n*-butyl phthalate (MnBP), mono-(3-carboxypropyl) phthalate (MCPP) and monobenzyl phthalate (MBzP) quartiles was significantly associated with reduced total hip or femur neck BMD. Subjects with the highest levels of MCPP phthalate, mono(carboxyoctyl) (MCOP) phthalate and the sum of three di(2-ethylhexyl) phthalate (ƩDEHP) metabolites were more likely to have an increased risk for total hip or femur neck osteoporosis than subjects with the lowest levels of these metabolites.	Increases in the urinary phthalate metabolites (except MCNP, MECPP, MEP and MiBP) were significantly associated with low BMD and high risk of osteoporosis in post-menopausal women. Phthalate exposure may adversely affect bone homeostasis and BMD in humans.
DeFlorio-Barker & Turyk (2016) [37]	Phthalates	*n* = 480, post-menopausal women from NHANES-survey (2005–2010).	A hypothesis-generating study, a cross-sectional study design.	BMD at the femoral neck and spine.	U-phthalate metabolites.	Mono-ethyl phthalate (MEP), molar sum of low molecular weight metabolites (mono-*n*-butyl phthalate (MNBP), mono-isobutyl phthalate (MIBP), MEP), molar sum of estrogenic metabolites (MNBP, MIBP, MEP, mono-benzyl phthalate (MBZP)) and an estrogenic equivalency factor were negatively associated with spinal BMD.	Due to the cross-sectional study design, uncertainty concerning the critical time window of exposure, the potential for exposure misclassification and residual confounding, no conclusions about association between phthalate metabolites and BMD in post-menopausal women could be drawn.
Khalil et al. (2016) [38]	PFASs	NHANES-survey (2009–2010): *n* = 1914, subjects of 12–80 years of age with BMD measurements for total femur (TFBMD), its sub-region femoral neck (FNBMD, *n* = 1914) and lumbar spine (LSBMD, *n* = 1605).	-	BMD (total femur, femoral neck, and lumbar spine) measured by DXA and physician-diagnosed osteoporosis.	Four PFASs (PFOA, PFOS, PFHxS, and PFNA) from blood serum.	Among men, higher serum PFAS (except for PFNA) concentrations were observed than among women (*p* < 0.001). In both sexes, serum PFOS concentrations were inversely associated with FNBMD (*p* < 0.05). Among women, significant negative associations between PFOS exposure and TFBMD and FNBMD and between PFOA exposure and TFBMD (*p* < 0.05) was observed. Among post-menopausal women, serum PFOS was negatively associated with TFBMD and FNBMD, and PFNA was negatively associated with TFBMD, FNBMD and LSBMD (all *p* < 0.05). One log unit increase in serum PFOA, PFHxS and PFNA increased osteoporosis prevalence in women as follows (aOR’s and 95% CI:s reported): 1.84 (1.17–2.905), 1.64 (1.14–2.38) and 1.45 (1.02–2.05), respectively. Among women, the prevalence of osteoporosis was significantly higher in the highest versus the lowest quartiles of PFOA, PFHxS and PFNA: 2.59 (1.01–6.67), 13.20 (2.72–64.15), and 3.23 (1.44–7.21), respectively.	Association between serum PFASs concentrations and lower BMD was observed; varied according to the specific PFAS and bone site assessed. Most of the associations were limited to women. In women, osteoporosis was associated with PFAS exposure.
Lin et al. (2014) [39]	PFASs	NHANES-survey (2005–2006, 2007–2008), *n* = 2339 (aged ≥ 20 years).	Cross-sectional design.	Total lumbar spine and total hip BMD measured by DXA and history of fractures.	The blood serum samples of PFOA and PFOS.	Among non-menopausal women, a 1-U increase in the natural log-transformed serum PFOS level was associated with a decrease in total lumbar spine BMD by 0.22 g/cm^2^ (95% CI 0.038–0.007, *p* = 0.006). No association was detected between PFOA and PFOS concentrations and femoral neck BMD or self-reported fractures.	Among non-menopausal women, a modest effect of serum PFOS concentration with decreased total lumbar spine BMD was observed.

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
