# Peer review of "Environmental Substances Associated with Osteoporosis–A Scoping Review"

_ijerph, 2021, doi:10.3390/ijerph18020738_

Round 1

Reviewer 1 Report

The scoping review by Elonheimo et al. provides an overview of current evidence on associations between select environmental factors and markers of osteoporosis (broadly defined). 

This reviewer recommends considerations of the below comments to improve data for the readership:

1) please justify the novelty of this work. Specifically, several original and review manuscripts have already been published on detrimental effects of evaluated environmental factors (e.g., heavy metals) on bone health. A rationale for the need to conduct this research would be plausible. 

2) please justify the set of exposure variables focused on in this work. For example, why Cd and Pb were selected but not arsenic and mercury, albeit the 2 latter factors have known adverse effects on bone health? Please clarify criteria/justification for the choice of some heavy metals, with phthalates and PFASs in the Introduction. 

3) Please provide clear and concise recommendations for future research to expand on Line 25 as it reads vaguely in the current form. 

4) please provide the most updated search date of retrieved records reviewed in this work.

Author Response

Thank you very much for your valuable comments. Please, find the responses below (highlighted in color red). 

Point 1: Please justify the novelty of this work. Specifically, several original and review manuscripts have already been published on detrimental effects of evaluated environmental factors (e.g., heavy metals) on bone health. A rationale for the need to conduct this research would be plausible. 

Response 1: Thank you for your well-pointed comment. A rationale for this scoping review derived from the idea to offer a disease-oriented starting point for the evaluation of the risks that environmental substances can cause on the human health. We wanted to make a distinction between the chemical-oriented approach and disease-oriented one, and rather scrutinize the diseases that could be (at least) partly caused due to the environmental factors. The rationale was also based on the HBM4EU project, where different diseases and related chemical exposure mechanisms are investigated. Rationale was elaborated on the lines 66-72 as follows: “The rationale of this review is to present a disease-oriented approach of the risks that environmental substances can cause on the human health, and more specifically on osteoporosis. Rather than focusing on the overall health effects of the specific substances, we chose to concentrate only on one specific health outcome, osteoporosis. This kind of approaching method can be informative when aiming to enhance public health and find ways to tackle the increasing burden of bone health disorders. According to our knowledge, this kind of scoping review has not been conducted earlier.”

Point 2: Please justify the set of exposure variables focused on in this work. For example, why Cd and Pb were selected but not arsenic and mercury, albeit the 2 latter factors have known adverse effects on bone health? Please clarify criteria/justification for the choice of some heavy metals, with phthalates and PFASs in the Introduction. 

Response 2: Thank you for pointing out an important issue and commenting on this. On the lines 101-105 you can kindly find the rationale on how the choosing of the chemicals was conducted. Briefly, according to the HBM4EU scoping documents and our literature search, we identified chemicals cadmium, lead, phthalates, and per- and polyfluoroalkyl substances to have an association with osteoporosis, and these substances were therefore included in our scoping review. As also mentioned on the lines 103-105, according to our search, the association was also suspected to exist for benzophenones (UV filters), bisphenols, PAHs and mercury. However, this evidence, according to our search, was not solid or consistent enough to be included in the review. Furthermore, an important reason for not selecting arsenic or mercury was that we wanted to solely focus on substances which showed the most impressive epidemiological evidence. This was the selection and exclusion method we decided to use. However, we realize that the selection could have been organized otherwise too, and we do understand your well-defined point. Therefore, in the future these other substances could be investigated as well, since, as you pointed out, arsenic and mercury are known to affect bone health adversely. According to our knowledge, more epidemiological evidence is called upon, since the majority of the studies regarding heavy metals are conducted on animals. The justification of our choice was clarified on the lines 61-64 as follows: “In this study, only the substances showing the strongest epidemiological evidence on humans according to current search are selected, and therefore e.g heavy metals such as arsenic and mercury are excluded.” Further clarification of the selection method can be found at the methods section (the lines 91-100).

Point 3: Please provide clear and concise recommendations for future research to expand on Line 25 as it reads vaguely in the current form. 

Response 3: Thank you again for a good comment. As the word count for the abstract is limited to 200 words maximum, we tried to be concise on these recommendations in this section. However, we elaborated that part in the abstract by adding to the lines 24-25 as follows: “Nevertheless, more epidemiological research on the relationship between health effects and the exposure to these chemicals is needed.” In the discussion part the future research need is more specifically defined (the lines 361-366). Given the word limit to the abstract (200 words max.), some minor changes were made in the abstract in order to adhere to the word limit. The biggest change was made on the lines 13-14 where sentence “Health-care services are dealing with significant costs due to osteoporosis” was omitted.

Point 4: Please provide the most updated search date of retrieved records reviewed in this work.

Response 4: Thank you for pointing that out. The most updated search date of retrieved records was provided and changes were made accordingly to the list of references. In the revised version the access dates are updated to the 17th of December (at the same time the links were double checked).

Reviewer 2 Report

This is a very interesting review examining the epidemiological studies of environmental substances associated with osteoporosis particularly focusing on the chemical substances’ cadmium, lead and PFASs. The authors have examined the evidence of published proceedings of population exposure to these substances and the correlation to low BMD. The manuscript is well written, and the authors have examined extensively a number of published articles  pertaining to this potential health problem and have given a good balanced overview of the potential correlation with the exposure of the chemicals with onset of bone diseases. One of the key issues in finding a causal relationship with exposure to these chemicals and increased levels of osteoporosis is to determine other osteoporotic key elements such as dietary deficiencies and change in hormone levels in patients.  Where there are high level exposures to chemicals such as Cadmium and the cause of Itai Itai disease then its considerably obvious of direct effects. The difficulty is trying to determine an effect to exposure to these chemicals at low levels over a long time on osteoporosis, and therefore implementing policy to reduce the levels will be quite hard to justify.

There are some minor questions to address.

Line 87 – 88 “Only epidemiological studies showing a possible connection between osteoporosis and the chemicals were included and opposing or controversial evidence was omitted from this review.” Not clear on this statement. Won’t epidemiological studies showing a possible connection between osteoporosis and the chemicals be controversial in its own right?

Line 207 -208 In the Wong study 2015 they demonstrated a link between increased levels of Pb and tibial bone becoming thinner and more fragile. The authors state that “Medical history including e.g. calcium and vitamin D intake, use of medication and incident fragility fractures was collected. Did this have any affect on the correlation of the levels of Pb with a thinner bone structure?

Author Response

Thank you very much for your valuable comments, and for taking the time to review our article. Please, find the responses below (highlighted in color red). 

Point 1: Line 87 – 88 “Only epidemiological studies showing a possible connection between osteoporosis and the chemicals were included and opposing or controversial evidence was omitted from this review.” Not clear on this statement. Won’t epidemiological studies showing a possible connection between osteoporosis and the chemicals be controversial in its own right?

Response 1: Thank you for your good point, which I totally agree with. Also the epidemiological studies selected for this scoping review are indeed controversial in findings. As mentioned in the discussion part of the text (the lines 359-360) the findings from the epidemiological studies selected are somewhat contradictory and only directional at the moment. Therefore the statement on the lines 96-98 was reformulated: “Only epidemiological studies showing a possible connection between osteoporosis and the chemicals were included, and if encountered, totally opposing evidence was omitted from this review.”

Point 2: Line 207 -208 In the Wong study 2015 they demonstrated a link between increased levels of Pb and tibial bone becoming thinner and more fragile. The authors state that “Medical history including e.g. calcium and vitamin D intake, use of medication and incident fragility fractures was collected. Did this have any affect on the correlation of the levels of Pb with a thinner bone structure?

Response 2: Thank you for your valuable comment. Wong et al. (2015) adjusted the results for age and BMI, diabetes or antiresorptive therapy. Higher amounts of bone Pb at the tibia were associated with thinner distal tibia cortices and integral volumetric BMD. A higher blood-to-bone lead partition coefficient was associated with larger trabecular separation, lower trabecular volumetric BMD, and trabecular number, per 100 µg Pb/g of bone mineral after adjusting for age and BMI, and remained significant after accounting for diabetes or use of antiresorptives. In the study in question, there was no mention, that intake of medication (other than antiresorptive therapy), using of supplements calcium or vitamin D or having prior fragility fractures would have affected on the correlations of the Pb-levels with bone structure or bone homeostasis. The information in question was collected from the CaMos-study (Canadian Multicenter Osteoporosis Study), and was used as background information to describe the bone characteristics of the participants. The statement on the lines 219-223 was reformulated: “Medical history including e.g. calcium and vitamin D intake, use of medication and incident fragility fractures was collected from the database of CaMos-study (Canadian Multicenter Osteoporosis Study), and it was used to describe the bone characteristics of the study subjects. This specific background information wasn`t applied in assessing the correlations of Pb-levels with the bone health.”

Round 2

Reviewer 1 Report

The manuscript has improved drastically after addressing this reviewer's comments and would be valuable to the readership of the Journal.